# DePT: Decomposed Prompt Tuning for Parameter-Efficient Fine-tuning

**Zhengxiang Shi, Aldo Lipani**
University College London, United Kingdom
{zhengxiang.shi.19,aldo.lipani}@ucl.ac.uk
https://github.com/ZhengxiangShi/DePT

## ABSTRACT

Prompt tuning (PT), where a small amount of trainable soft (continuous) prompt vectors is affixed to the model input, has shown promising results across various tasks and model architecture for parameter-efficient fine-tuning (PEFT). PT stands out from other PEFT approaches because it maintains competitive performance with fewer trainable parameters and does not drastically scale up its parameters as the model size expands. However, PT introduces extra soft prompt tokens, leading to longer input sequences, which significantly impacts training/inference time and memory usage due to the Transformer's quadratic complexity. Particularly concerning for Large Language Models (LLMs) that face heavy daily querying. To address this issue, we propose **De**composed **P**rompt **T**uning (DePT), which decomposes the soft prompt into a shorter soft prompt and a pair of low-rank matrices that are then optimised with two different learning rates. This allows DePT to achieve better performance while saving substantial memory and time costs compared to vanilla PT and its variants, without changing trainable parameter sizes. Through extensive experiments on 23 natural language processing (NLP) and vision-language (VL) tasks, we demonstrate that DePT outperforms state-of-the-art PEFT approaches, including the full fine-tuning baseline, in some scenarios. Additionally, we empirically show that DePT grows more efficient as the model size increases. Our further study reveals that DePT integrates seamlessly with parameter-efficient transfer learning in the few-shot learning setting and highlights its adaptability to various model architectures and sizes.

## 1 INTRODUCTION

Fine-tuning (FT) language models (LMs) (Raffel et al., 2020; Touvron et al., 2023) on downstream tasks offers large performance improvements across various natural language processing (NLP) tasks, but it requires updating and storing full parameters of the LMs (see Figure 1a), which is especially expensive when LMs contain hundreds of millions or even billions of parameters. Prompt engineering (Brown et al., 2020) does not update any parameters while it is typically hard to design and has a high-performance variance (Wang et al., 2023a) (see Figure 1c). Consequently, parameter-efficient fine-tuning (PEFT) approaches (Liu et al., 2022) have attracted growing interest, aiming to learn only a small number of parameters per task while maintaining performance levels comparable to full fine-tuning.

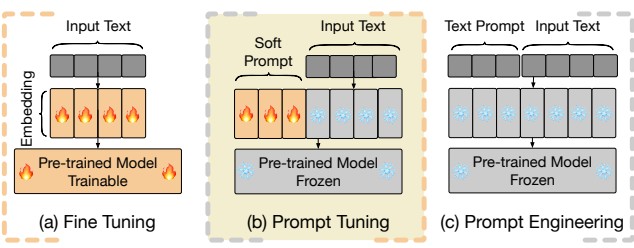

Figure 1: The overview of Fine Tuning (FT), Prompt Tuning (PT), and Prompting Engineering. PT increases the length of the input sequence, leading to much greater computational demands during train and inference phrases.

Prompt Tuning (PT) (Lester et al., 2021) has emerged as a promising PEFT approach, which appends trainable continuous prompt vectors to the input (see Figure 1b). PT stands out from other PEFT approaches as it maintains competitive performance with fewer trainable parameters and does not drastically scale up its trainable parameters as the model size expands. Recent works suggest that the majority of the LM's knowledge is acquired during its pretraining phase (Zhou et al., 2023), and

that in-context learning (ICL) with just a few carefully designed stylistic examples and a carefully designed system prompt can achieve impressive alignment results (Lin et al., 2023). Considering scenarios where tasks have already been somewhat understood by LMs and the key challenge is just to properly prompt the LMs, PT emerges as a potentially better option to other PEFT approaches.

While PT has shown promising results across various tasks and models, it has two major limitations: (1) PT often suffers from slow convergence and is sensitive to the initialization (Lester et al., 2021; Vu et al., 2022; Wang et al., 2023b); and (2) PT extends the total length of the input sequence, consequently exacerbating the computation demand (*i.e.,* train/inference time and memory cost), due to the quadratic complexity of the Transformer (Vaswani et al., 2017). This is further accentuated given the slow convergence issue. Recent studies (Su et al., 2022; Vu et al., 2022; Li et al., 2022) have proposed the variants of the vanilla PT to tackle the first issue by initially pre-training soft prompts on a variety of source tasks, which is known as *Parameter-Efficient Transfer Learning* (PETL), as depicted in Figure 2a. Some studies (Asai et al., 2022; Wang et al., 2023b) also improve the performance of the PT by jointly training learned prompts from these source tasks on multiple target tasks (referred to as *Multi-task Learning*). However, the issue of increased computational load due to the extension of sequence length remains largely unaddressed. While PETL approaches can reduce the training steps for model convergence, each optimization step remains computationally expensive in terms of time and memory. Most importantly, it does not enhance the efficiency during the inference phase, which is particularly crucial in the era of Large Language Models (LLMs), considering that the trained models may be queried millions of times per day.

In this work, we propose **De**composed **P**rompt **T**uning (DEPT), which decomposes a trainable soft prompt into a shorter soft prompt and a couple of low-rank matrices, where the multiplication of low-rank matrices is then added element-wise to frozen word embeddings, as shown in Figure 2b (§2.2). This shorter soft prompt and the updated word embedding matrix are then optimised using two different learning rates - a crucial step for model convergence (§3.4). The intuition of this design is to enable representation updates within the frozen word embedding, thereby increasing the adaptability of input representations that were previously unavailable. Experimental results on 23 natural language processing (NLP) and vision-language (VL) tasks demonstrate DEPT outperforms the state-of-the-art PEFT

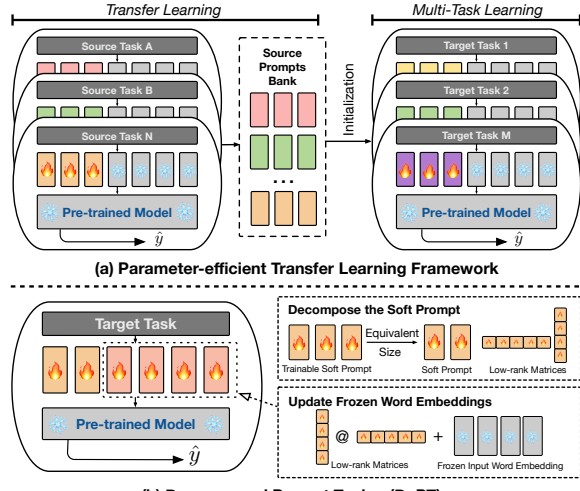

Figure 2: The overview of the PETL framework (***Top***) and our method DEPT (***Bottom***). DEPT decomposes a trainable soft prompt of the vanilla PT into a shorter soft prompt and a couple of low-rank matrices, where the multiplication of low-rank matrices serves to update frozen word embedding.

approaches, including the full fine-tuning baseline in certain scenarios (§3.2). Our study empirically shows that DEPT largely improves the training efficiency across various model architectures and sizes, saving more than 20% (using T5-BASE) in both training time and memory costs compared to the vanilla PT. Importantly, DEPT becomes increasingly efficient as the model size grows, making it particularly advantageous and suitable for LLMs (§3.3). Furthermore, our additional analysis in the few-shot learning setting reveals the DEPT's compatibility with PETL approaches (§3.4).

In summary, the main contributions of this paper are as follows:

- We propose DEPT method, which addresses a key efficiency limitation of Prompt Tuning by decomposing its soft prompt to reduce input sequence length. DEPT largely improves the training and inference efficiency, in terms of both time and memory costs;

- Our comprehensive evaluation on 23 NLP and VL tasks demonstrates that DEPT outperforms state-of-the-art PEFT approaches, including the full fine-tuning in some scenarios.

Additionally, our experiments show that DEPT smoothly integrates with PETL approaches and the advantage of DEPT persists in the few-shot learning setting;

- We empirically show that DEPT becomes increasingly efficient as the model size grows, making it particularly well-suited for LLMs. Furthermore, DEPT is orthogonal to various PEFT approaches (*i.e.,* Adapter, LoRA) and can be easily combined together.

## 2 METHOD

In this section, we first revisit background of Prompt Tuning (PT) in §2.1 and then introduce our proposed method, **De**composed **P**rompt **T**uning (DEPT) in §2.2.

### 2.1 BACKGROUND: PROMPT TUNING (PT)

Let $L \triangleq \{\boldsymbol{x}_i, \boldsymbol{y}_i\}_{i=1}^N$ denote $N$ labelled training data for the target task $\mathcal{T}$. Given a backbone model parameterised by $\Theta$, each input text $\boldsymbol{x}_i$ is mapped into a sequence of word embeddings $\boldsymbol{W}_i \in \mathbb{R}^{s \times d}$, where $s$ and $d$ represent the maximum sequence length and the dimension of word embeddings. PT appends a trainable prompt matrix $\boldsymbol{P} \in \mathbb{R}^{l \times d}$ to the frozen word embedding matrix $\boldsymbol{W}_i$, where $l$ is a hyper-parameter for the number of virtual tokens. The soft prompt $\boldsymbol{P}$ can be initialised either randomly or by sampling word embeddings from the vocabulary. Consequently, the model's input becomes the combined matrix $[\boldsymbol{P}; \boldsymbol{W}_i] \in \mathbb{R}^{(l+s) \times d}$. The targeted loss function is formulated as:

$$\mathcal{L}_{\mathrm{PT}} = - \sum_i \log P(\boldsymbol{y}_i \,|\, [\boldsymbol{P}, \boldsymbol{W}_i]\,; \Theta), \tag{1}$$

where the loss function is only optimised with respect to the soft prompt matrix $\boldsymbol{P}$.

### 2.2 OUR APPROACH: DECOMPOSED PROMPT TUNING (DEPT)

**The decomposition of the soft prompt.** DEPT differs from the vanilla PT method in the aspect of inputs. As shown in Figure 2b, we decompose a trainable prompt matrix $\boldsymbol{P} \in \mathbb{R}^{l \times d}$ from the vanilla PT into two components: (1) a shorter trainable prompt matrix $\boldsymbol{P}_s \in \mathbb{R}^{m \times d}$; and (2) a pair of low-rank matrices, $\boldsymbol{A} \in \mathbb{R}^{s \times r}$ and $\boldsymbol{B} \in \mathbb{R}^{r \times d}$, where typically the rank of the matrices $r \ll \min(s, d)$. The first component, the smaller trainable prompt matrix, is appended to the word embedding matrix in a similar manner as in the vanilla PT. The second component uses the multiplication of two low-rank matrices to represent the update of the word embedding through a coordinate-wise sum:

$$\boldsymbol{W}_i^{'} = \boldsymbol{W}_i + \Delta \boldsymbol{W}_i = \boldsymbol{W}_i + \boldsymbol{B}\boldsymbol{A} \in \mathbb{R}^{s \times d}, \tag{2}$$

where $\boldsymbol{W}_i$ is frozen and does not receive gradient updates during the training, whereas $\boldsymbol{A}$ and $\boldsymbol{B}$ are trainable. Following Hu et al. (2021), we use a random Gaussian initialization for $A$ and zero for $B$, so $\Delta W = BA$ is zero when the training starts. The loss function is then optimised as follows:

$$\mathcal{L}_{\mathrm{DEPT}} = - \sum_i \log P(\boldsymbol{y}_i \,|\, [\boldsymbol{P}_s, \boldsymbol{W}_i^{'}]\,; \Theta) \tag{3}$$

In our experiment, we choose the values of $m$ and $r$ to satisfy the equation $l \times d = m \times d + (s + d) \times r$ for maintaining the exact size of trainable parameters as in the vanilla PT. Consequently, $m$ is always less than $l$ when $r > 0$. This design improves memory efficiency and reduces computational expense compared to the vanilla PT, as the shorter input sequence length (*i.e.,* $m + s < l + s$) substantially reduces computation due to the quadratic complexity of the Transformer (Vaswani et al., 2017).

**Two rates of learning.** DEPT also differs from the vanilla PT in training. We train the shorter trainable prompt matrix, $\boldsymbol{P}_s$, with the learning rate $\alpha_1$ and the pair of low-rank matrices, $\boldsymbol{A}$ and $\boldsymbol{B}$, with the learning rate $\alpha_2$, rather than use a single learning rate as in the vanilla PT. The $\alpha_1$ is typically much larger than the $\alpha_2$. We will empirically validate the importance of this choice in §3.4. However, DEPT may introduces extra training costs for the hyperparameter optimization (see §5).

## 3 EXPERIMENTS AND RESULTS

In this section, we introduce our experimental setup (see §3.1), evaluate the performance of DEPT across 23 different NLP and VL tasks (see §3.2), and assess relative train/inference time and memory cost of DEPT (see §3.3), and explore the effectiveness of DEPT in the few-shot learning setting and importance of two different learning rates for training DEPT (see §3.4).

## 3.1 EXPERIMENTAL SETUP

**Datasets and tasks.** We evaluate our proposed method DEPT on 21 NLP tasks and 2 vision-language tasks. For NLP tasks, we follow the previous works (Vu et al., 2022; Sung et al., 2022b; Asai et al., 2022; Wang et al., 2023b) and use various datasets sourced from: (1) GLUE (Wang et al., 2018) benchmark, including MNLI (Williams et al., 2018), QQP[1], QNLI (Rajpurkar et al., 2016), SST-2 (Socher et al., 2013), STS-B (Cer et al., 2017), MRPC (Dolan & Brockett, 2005), RTE (Giampiccolo et al., 2007) and CoLA (Warstadt et al., 2019); (2) SuperGLUE benchmark (Wang et al., 2019), including MultiRC (Khashabi et al., 2018), BoolQ (Clark et al., 2019), WiC (Pilehvar & Camacho-Collados, 2019), WSC (Levesque et al., 2012), and CB (De Marneffe et al., 2019); (3) MRQA 2019 Shared Task (Fisch et al., 2019), including Natural Questions (Kwiatkowski et al., 2019), HotpotQA (Yang et al., 2018), SearchQA (Dunn et al., 2017) and NewsQA (Trischler et al., 2017); (4) other datasets, including WinoGrande (Sakaguchi et al., 2021), Yelp-2 (Zhang et al., 2015), SciTail (Khot et al., 2018) and PAWS-Wiki (Zhang et al., 2019). For vision-language tasks, we follow prior works (Sung et al., 2022a;b) to experiment with the visual question-answering task, VQA (Goyal et al., 2017), and the image caption generation task, MSCOCO (Chen et al., 2015).

**Baselines.** We compare DEPT with a variety of baselines: (1) fine-tuning (FT), where all the model parameters are tuned during adaptation on each downstream task; (2) the vanilla PT (Lester et al., 2021), where target prompt vectors are initialized by randomly sampled top vocabularies, and its variants using additional transfer and multi-task learning, including SPoT (Vu et al., 2022), AT-TEMPT (Asai et al., 2022), and MPT (Wang et al., 2023b); (3) state-of-the-art PEFT approaches including Adapters (Houlsby et al., 2019), AdapterDrop (Rücklé et al., 2021), BitFit (Ben Zaken et al., 2022), HyperFomer (Karimi Mahabadi et al., 2021), HyperDecoder (Ivison & Peters, 2022), P-tuning (Liu et al., 2021), LoRA (Hu et al., 2021), LST (Sung et al., 2022b), and their multi-task learning variants. For a fair comparison, we directly quote performance metrics from published papers (Mahabadi et al., 2021; Karimi Mahabadi et al., 2021; Asai et al., 2022; Wang et al., 2023b; Sung et al., 2022b) for a fair comparison, where all these baselines using the T5-BASE as the backbone and adhere to the train, validation and test splits used by Karimi Mahabadi et al. (2021); Mahabadi et al. (2021) for NLP tasks and by Sung et al. (2022b) for vision-language tasks.

**Implementation details.** In our study, we mainly experiment using the T5-BASE model with 220M parameters (Raffel et al., 2020). We consistently set the number of virtual tokens $l$ as 100 across all tasks for the vanilla PT and adjust the hyper-parameters of DEPT accordingly to maintain the equivalent number of trainable parameters. For instance, the vanilla PT contains $l \times d$ trainable parameters where the hidden size $d$ is 768 for the T5-BASE, and DEPT can configure the number of virtual tokens $m$ as 40 and the rank of low matrices $r$ as 45, resulting in $m \times d + (s + d) \times r$ trainable parameters. This yields a total of $76,800$ trainable parameters, aligning with the vanilla PT. For VL tasks, we utilise the CLIP-T5 architecture which combines CLIP (Radford et al., 2021) and T5-BASE (Raffel et al., 2020), with the CLIP frozen. We follow the prior work (Sung et al., 2022b) to concatenate the visual representation from CLIP with the text embedding from the T5-BASE, where a trainable visual projection layer is used between CLIP and T5 to align the visual representation to the same dimension as the text embedding.

We also extend our evaluation to include T5-SMALL (60M), T5-LARGE (770M), GPT2-SMALL (110M), GPT2-MEDIUM (345M), and GPT2-LARGE (774M) models. In the few-shot experiments, we randomly select $k$ examples three times from the training set and report the mean and standard deviations for each $k$-shot experiment. Following the prior works in PETL for PT (Vu et al., 2022; Su et al., 2022; Asai et al., 2022), we use MNLI, QQP, SST-2, SQUAD (Rajpurkar et al., 2016), and ReCoRD (Zhang et al., 2018) as five source tasks. Our soft prompt and low-rank matrix pairs are initialized from the soft prompts derived from one of these selected source tasks. Please see more hyper-parameter and implementation details in Appendix §D.

## 3.2 MAIN RESULTS

This section shows the empirical evidence supporting the effectiveness of our proposed method DEPT across 23 NLP and VL tasks. Table 1, 2, and 3 present our experimental results on GLUE and SuperGLUE benchmarks, MRQA 2019 Shared Task and four other NLP datasets, as well as two VL tasks. Additionally, we visualise the model performance against the number of trainable

---

[1]https://www.quora.com/q/quoradata/

Table 1: Test results on GLUE and SuperGLUE benchmarks, with the corresponding size of trainable parameters. All of the results are based on T5-BASE models. We use Pearson correlation for STS-B, F1 for MultiRC (Multi), and accuracy for other tasks as evaluation metrics.

| Method | #Para | GLUE | | | | | | | | | SuperGLUE | | | | | |
|---|---|---|---|---|---|---|---|---|---|---|---|---|---|---|---|---|
| | | MNLI | QQP | QNLI | SST-2 | STS-B | MRPC | RTE | CoLA | Mean | Multi | Bool | WiC | WSC | CB | Mean |
| *Single-Task Learning* | | | | | | | | | | | | | | | | |
| Fine-tuning[1] | 220M | 86.8 | 91.6 | 93.0 | 94.6 | 89.7 | 90.2 | 71.9 | 61.8 | 84.9 | 72.8 | 81.1 | 70.2 | 59.6 | 85.7 | 73.9 |
| Adapter[1] | 1.9M | 86.5 | 90.2 | 93.2 | 93.8 | 90.7 | 85.3 | 71.9 | 64.0 | 84.5 | 75.9 | 82.5 | 67.1 | 67.3 | 85.7 | 75.7 |
| AdapterDrop[1] | 1.1M | 86.3 | 90.2 | 93.2 | 93.6 | 91.4 | 86.3 | 71.2 | 62.7 | 84.4 | 72.9 | 82.3 | 68.3 | 67.3 | 85.7 | 75.3 |
| BitFit[1] | 280k | 85.3 | 90.1 | 93.0 | 94.2 | 90.9 | 86.8 | 67.6 | 58.2 | 83.3 | 74.5 | 79.6 | 70.0 | 59.6 | 78.6 | 72.5 |
| LoRA[2] | 3.8M | 86.3 | 89.0 | 93.2 | 94.3 | 90.9 | 90.1 | 75.5 | 63.3 | 85.3 | 72.6 | 81.3 | 68.3 | 67.3 | 92.9 | 76.5 |
| LST[2] | 3.8M | 85.6 | 88.8 | 93.3 | 94.1 | 90.7 | 90.4 | 71.9 | 58.1 | 84.1 | – | – | – | – | – | – |
| PT[4] | 76.8k | 83.4 | 90.2 | 93.1 | 91.9 | 90.2 | 90.1 | 78.8 | 60.7 | 84.8 | 65.7 | 63.7 | 50.8 | 51.9 | 67.9 | 60.0 |
| DEPT (ours) | 76.8k | 85.0 | 90.4 | 93.2 | 94.2 | 90.8 | 90.7 | 79.1 | 63.8 | 85.9 | 74.3 | 79.3 | 68.7 | 67.3 | 92.9 | 76.5 |
| *Multi-task Learning* | | | | | | | | | | | | | | | | |
| Fine-tuning (m)[1] | 28M | 85.7 | 91.1 | 92.0 | 92.5 | 88.8 | 90.2 | 75.4 | 54.9 | 83.8 | 74.4 | 81.1 | 70.0 | 71.2 | 85.7 | 76.1 |
| Adapter (m)[1] | 1.8M | 86.3 | 90.5 | 93.2 | 93.0 | 89.9 | 90.2 | 70.3 | 61.5 | 84.4 | 72.6 | 82.3 | 66.5 | 67.3 | 89.3 | 75.6 |
| HyperFormer (m)[1] | 638k | 85.7 | 90.0 | 93.0 | 94.0 | 89.7 | 87.2 | 75.4 | 63.7 | 84.8 | 72.9 | 82.5 | 69.0 | 67.3 | 85.7 | 75.4 |
| HyperDecoder (m)[1] | 1.8M | 86.0 | 90.5 | 93.4 | 94.0 | 90.5 | 87.7 | 71.7 | 55.9 | 83.7 | 70.4 | 78.8 | 67.1 | 61.5 | 82.1 | 72.0 |
| *Single-Task Training + Transfer Learning* | | | | | | | | | | | | | | | | |
| SPoT[1] | 76.8k | 85.4 | 90.1 | 93.0 | 93.4 | 90.0 | 79.7 | 69.8 | 57.1 | 82.3 | 74.0 | 77.2 | 67.0 | 50.0 | 46.4 | 62.9 |
| ATTEMPT[1] | 232k | 84.3 | 90.3 | 93.0 | 93.2 | 89.7 | 85.7 | 73.4 | 57.4 | 83.4 | 74.4 | 78.8 | 66.8 | 53.8 | 78.6 | 70.5 |
| MPT[3] | 77.6k | 85.9 | 90.3 | 93.1 | 93.8 | 90.4 | 89.1 | 79.4 | 62.4 | 85.6 | 74.8 | 79.6 | 69.0 | 67.3 | 79.8 | 74.1 |
| *Multi-task Learning + Transfer Learning* | | | | | | | | | | | | | | | | |
| ATTEMPT (m)[3] | 96k* | 83.8 | 90.0 | 93.1 | 93.7 | 90.8 | 86.1 | 79.9 | 64.3 | 85.2 | 74.4 | 78.5 | 66.5 | 69.2 | 82.1 | 74.1 |
| MPT (m)[3] | 10.5k* | 84.3 | 90.0 | 93.0 | 93.3 | 90.4 | 89.2 | 82.7 | 63.5 | 85.8 | 74.8 | 79.2 | 70.2 | 67.3 | 89.3 | 76.1 |

[1] sourced from Asai et al. (2022). [2] sourced from Sung et al. (2022b). [3] sourced from Wang et al. (2023b). [4] we reproduce and substantially increase the performance of the vanilla PT reported in the prior work (Asai et al., 2022). * These values are obtained after amortizing over 8 tasks, and the minimal number of parameters to perform a single task remains 232k and 77.6k for ATTEMPT and MPT. (m) represents additional multi-task training.

parameters for GLUE and SuperGLUE in Figure 6 of Appendix §A. Furthermore, we evaluate the performance of DEPT using LLAMA-2 (Touvron et al., 2023) in Appendix §B. Experimental results reveal three key findings: (1) DEPT consistently outperforms the vanilla PT and its PETL variants; (2) DEPT achieves competitive or even better performance than state-of-the-art PEFT approaches while using fewer trainable parameters; and (3) DEPT falls short in some certain tasks. Below we delve deeper with respect to various tasks.

**#1. Performance on GLUE and SuperGLUE benchmarks.** As shown in Table 1, our experimental result indicates that DEPT outperforms state-of-the-art PEFT approaches, such as Adapter, LoRA and LST on the GLUE and SuperGLUE benchmarks, while using fewer trainable parameters. Remarkably, DEPT also outperforms the full fine-tuning baseline on both benchmarks. In addition, DEPT outperforms vanilla PT and all the variants of PT that introduce additional transfer learning and multi-task learning. For example, ATTEMPT, which requires additional training for the soft prompt on the source tasks, achieves an average score of 83.4 on the GLUE benchmark and 70.5 on the SuperGLUE benchmark. Meanwhile, DEPT outperforms ATTEMPT with scores of 85.9 and 76.5 on GLUE and SuperGLUE, despite training fewer parameters. Similarly, DEPT surpasses MPT with 0.1% on the GLUE benchmark and 0.4% on the SuperGLUE benchmark, without utilizing additional transfer learning or multi-task learning. These results are achieved with less inference time and reduced memory resources (refer to §3.3 for specifics), which validates the effectiveness of DEPT. As the PT often underperforms in scenarios with limited labelled data (Gu et al., 2022), we investigate the compatibility of DEPT and PETL later in the few-shot learning setting (§3.4).

**#2. Performance on MRQA 2019 Shared Task and other NLP datasets.** Table 2 presents the performance of various PEFT approaches, including DEPT, on the MRQA 2019 Shared Task and four other datasets. We observe that DEPT improves the average performance of the vanilla PT by a substantial margin of +3.6% on MRQA and +14.2% on the other datasets. DEPT exceeds the performance of the PT variants that leverage additional transfer and multi-task learning, without introducing extra trainable parameters to the vanilla PT or relying on any PETL approaches. While

Table 2: Test results on MRQA 2019 Shared Task and other datasets using the T5-BASE model. We report the $F_1$ for MRQA tasks and accuracy for other datasets across three seeds, with standard deviations in subscripts. All baseline results are directly quoted from Wang et al. (2023b).

| Method | #Para | MRQA | | | | | Others | | | | |
|---|---|---|---|---|---|---|---|---|---|---|---|
| | | NQ | HP | SQA | News | Mean | WG | Yelp | SciTail | PAWS | Mean |
| Fine Tuning | 220M | 75.1 | 77.5 | 81.1 | 65.2 | 74.7 | 61.9 | 96.7 | 95.8 | 94.1 | 87.1 |
| Adapters | 1.9M | 74.2 | 77.6 | 81.4 | 65.6 | 74.7 | 59.2 | 96.9 | 94.5 | 94.3 | 86.2 |
| BitFit | 280K | 70.7 | 75.5 | 77.7 | 64.1 | 72.0 | 57.2 | 94.7 | 94.7 | 92.0 | 84.7 |
| LoRA | 3.8M | 72.4 | 62.3 | 72.5 | 56.9 | 66.0 | 58.2 | 97.1 | 94.7 | 94.0 | 86.0 |
| PT | 76.8K | 67.9 | 72.9 | 75.7 | 61.1 | 69.4 | 49.6 | 95.1 | 87.9 | 55.8 | 72.1 |
| SPoT | 76.8K | 68.2 | 74.8 | 75.3 | 58.2 | 69.1 | 50.4 | 95.4 | 91.2 | 91.1 | 82.0 |
| ATTEMPT | 232K | 70.4 | 75.2 | 77.3 | 62.8 | 71.4 | 57.6 | 96.7 | 93.1 | 92.1 | 84.9 |
| MPT | 77.6K | $72.0_{0.1}$ | $75.8_{0.1}$ | $77.2_{0.1}$ | $63.7_{0.1}$ | 72.2 | $56.5_{0.9}$ | $96.4_{0.0}$ | $95.5_{0.1}$ | $93.5_{0.1}$ | 85.5 |
| DEPT (ours) | 76.8K | $73.2_{0.1}$ | $76.8_{0.3}$ | $77.6_{0.2}$ | $64.4_{0.1}$ | 73.0 | $59.0_{0.2}$ | $96.8_{0.1}$ | $95.6_{0.2}$ | $93.7_{0.1}$ | 86.3 |

DEPT improves over the vanilla PT and its variants are promising, there remains a disparity in performance when compared to the full fine-tuning baseline. Investigating ways to incorporate DEPT with other PEFT methods, such as LoRA and Adapter, may provide a valuable direction for future research towards narrowing this performance gap.

**#3. Performance on Vision-Language tasks.** Table 3 provides an overview of the performance of various PEFT approaches on two VL tasks, specifically VQA and MS COCO Caption Generation. Results show that DEPT, while updating much fewer parameters, achieves a CIDEr score of 113.7 on the MS COCO Caption Generation task, outperforming state-of-the-art PEFT approaches. This suggests the effectiveness of our proposed method. However, while DEPT outperforms methods such as P-tuning and BitFit on the VQA dataset, it still falls short of the full fine-tuning performance. This suggests that in certain tasks, the use of a greater number of trainable parameters could be beneficial.

Table 3: Test results on the VQA and MSCOCO dataset using T5-BASE model. We report average results across three seeds, with standard deviations in subscripts. All baseline results are directly quoted from Sung et al. (2022b). The best performance for each column is highlighted in blue.

| Method | Updated Params (%) | VQA Karpathy test Acc. (%) | MSCOCO Karpathy test CIDEr |
|---|---|---|---|
| FT | 100 | $67.1_{0.1}$ | $112.2_{0.3}$ |
| Adapters | 7.98 | $67.1_{0.1}$ | $111.8_{0.1}$ |
| LoRA | 7.54 | $63.7_{0.2}$ | $110.3_{0.4}$ |
| BitFit | 0.83 | $55.1_{0.2}$ | $101.2_{0.2}$ |
| P-Tuning | 1.26 | $47.4_{0.7}$ | $96.1_{0.9}$ |
| LST | 7.46 | $66.5_{0.1}$ | $113.5_{0.3}$ |
| DEPT (ours) | 0.74 | $59.8_{0.4}$ | $113.7_{0.3}$ |

### 3.3 TIME AND MEMORY EFFICIENCY

This section shows the empirical evidence supporting the efficiency of DEPT, spanning over diverse model architectures of varying scales on the GLUE benchmark. To ensure a fair comparison, we consistently keep the number of trainable parameters in DEPT the same as that in the vanilla PT ($l = 100$). As a result, once we choose the length of the soft prompt $m$ in DEPT, the rank of the low-rank matrices $r$ becomes determined. In our experiments, we primarily compare DEPT with the vanilla PT using 5 different lengths of soft prompt $m$ (*i.e.,* 0, 20, 40, 60, 80). Figure 3 and 4 depict the average GLUE performance of DEPT, along with the associated training/inference time and memory cost compared to the vanilla PT. Below we discuss two key findings.

**# 1. DEPT improves time and memory efficiency substantially.** Figure 3 presents the mean performance of DEPT, associated with average training time and memory, on the GLUE benchmarks, against different lengths of soft prompt $m$. The average training time and memory costs are computed across 8 tasks on the GLUE benchmark and three different model sizes. Both the encoder-decoder (T5) and decoder-only (GPT-2) models are evaluated across three different model sizes. The study reveals that decomposing the soft prompt ($l = 100$) into a small soft prompt and low-rank matrices delivers comparable or even better performance while substantially enhancing the efficiency of training and reducing memory utilization. Specifically, using a soft prompt length greater than 20 in DEPT with the T5 model leads to a better average performance on the GLUE benchmark to vanilla PT, while improving the efficiency of training and reducing memory utilization by approximately 25%. This improvement is more pronounced (37% on the SST-2 dataset) when we test DEPT (with $m = 60$) using the T5-3B model (see §B for details). Similar observations are also found when the GPT model is used, suggesting the adaptability of DEPT for different model architectures. It is worth noting that DEPT may have a notable performance drop regardless

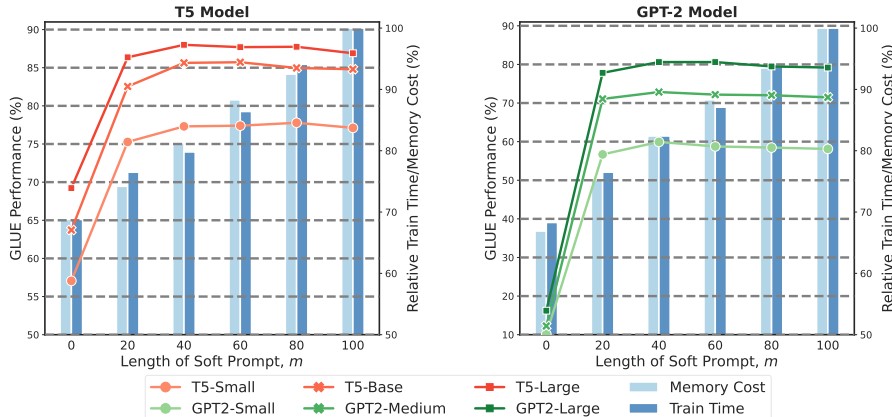

Figure 3: Performance on the GLUE benchmark for different soft prompt lengths $m$ in DEPT, associated with corresponding relative train time and memory cost. The time and memory are averaged over different model sizes using batch size as 16. DEPT consistently uses the same number of trainable parameters as the vanilla PT ($m$=100).

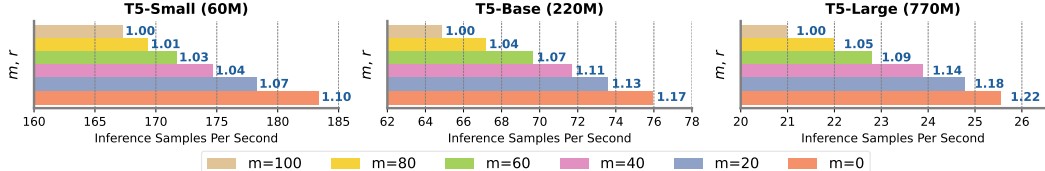

Figure 4: Average inference speed on GLUE benchmark using varying soft prompt length $m$ and the rank of low-rank matrices $r$, keeping the total number of trainable parameters constant. Small texts in blue indicate the speed relative to the vanilla PT (represented by brown) ($m$=100).

of using T5 or GPT-2, when the soft prompt is eliminated ($m = 0$) and the model solely depends on the pair of low-rank matrices.

**# 2. DEPT grows more efficient as the model size increases.** Figure 4 represents the inference speed, measured by the average number of samples evaluated per second on the GLUE benchmark using a single RTX 3090 GPU. The inference time is computed using the Huggingface Trainer Class. We observe that the relative improvement in the number of inference samples per second over vanilla PT grows as the model size increases. For example, when using the T5-SMALL model, the vanilla PT evaluates 167.3 samples per second, while DEPT ($m = 20$) evaluates 178.3 samples per second, resulting in a 6.5% boost in inference speed. In contrast, when the T5-LARGE is utilized, the vanilla PT evaluates 21.0 samples per second and DEPT ($m = 20$) evaluates 24.8 samples per second, resulting in an 18.1% increase in inference speed, a substantial rise from the previous 6.5%. This indicates that DEPT is particularly beneficial and more applicable in the context of LLMs. Please refer to Appendix §B for the inference speed of DEPT and PT using T5-3B and LLAMA-2.

## 3.4 FURTHER ANALYSIS

**Few-shot Learning.** The vanilla PT often underperforms in the few-shot learning tasks (Gu et al., 2022) due to the first limitation discussed in §1. To evaluate the performance of DEPT in the few-shot setting, we employ the transfer learning method inspired by the recent PETL studies, as illustrated in Figure 2a. Specifically, we pre-train both the soft prompt and the low-rank pair on source tasks and select the best checkpoint before proceeding with the target task. Following prior works (Karimi Mahabadi et al., 2021; Asai et al., 2022; Wang et al., 2023b), we evaluate the effectiveness of DEPT across 14 NLP tasks, with $k$ training examples where k = 4, 16, 32. Our experimental findings reveal two key observations as follows: (1) DEPT integrates seamlessly with PETL approaches; and (2) DEPT attains competitive or even better performance than state-of-the-art PEFT approaches in the few-shot learning setting.

Table 4 compares the effectiveness of our proposed method DEPT with various PEFT approaches in few-shot experiments, including full fine-tuning (FT), Adapters (AD), vanilla PT (PT), SPoT (ST),

Table 4: Few-shot learning results with $k = \{4, 16, 32\}$ on the SuperGLUE BooQ, SuperGLUE CB and SciTail datasets. We report average results across three seeds, with standard deviations in subscripts. Baseline results are directly quoted from Wang et al. (2023b). The best performance for each row is highlighted in blue.

| Task | $k$-shot #Para | FT 220M | AD 1.9M | PT 76.8K | ST 76.8K | HF 638K | $(IA)^3$ 55.3K | ATP 232K | MPT 77.6K | DEPT 76.8K |
|------|------|------|------|------|------|------|------|------|------|------|
| BoolQ | 4 | 50.5 | 53.4 | 61.6 | 50.5 | 48.0 | 56.7 | 61.8 | 62.2 | $62.7_{5.4}$ |
|  | 16 | 56.5 | 51.4 | 61.9 | 50.6 | 50.2 | 62.0 | 60.0 | 63.3 | $66.9_{4.4}$ |
|  | 32 | 58.4 | 54.5 | 61.7 | 61.2 | 58.3 | 67.2 | 65.3 | 68.9 | $67.2_{3.4}$ |
| CB | 4 | 57.7 | 51.1 | 53.5 | 71.4 | 60.7 | 65.5 | 67.9 | 73.6 | $75.0_{5.1}$ |
|  | 16 | 77.0 | 74.8 | 63.5 | 64.3 | 76.3 | 71.4 | 71.4 | 78.6 | $78.6_{4.3}$ |
|  | 32 | 80.0 | 74.8 | 67.8 | 64.3 | 81.4 | 75.0 | 78.5 | 82.1 | $82.1_{2.3}$ |
| SciTail | 4 | 79.6 | 79.5 | 57.7 | 69.6 | 82.0 | 65.4 | 80.2 | 80.2 | $78.1_{2.5}$ |
|  | 16 | 80.0 | 83.2 | 60.8 | 71.9 | 86.5 | 74.4 | 79.5 | 87.3 | $78.5_{1.4}$ |
|  | 32 | 81.9 | 85.0 | 60.2 | 71.9 | 85.8 | 80.4 | 80.2 | 86.3 | $85.4_{3.1}$ |

Table 5: Few-shot learning results with $k = \{4, 16, 32\}$ on GLUE and SuperGLUE benchmarks. We report average results across three seeds, with standard deviations in subscripts. Baseline results are directly quoted from Wang et al. (2023b).

| $k$-shot | Method | GLUE | | | | | | | | | SuperGLUE | | | | | |
|------|------|------|------|------|------|------|------|------|------|------|------|------|------|------|------|------|
|  |  | MNLI | QQP | QNLI | SST-2 | STS-B | MRPC | RTE | CoLA | Avg. | Multi | BoolQ | WiC | WSC | CB | Avg. |
| 4 | PT | 40.1 | 63.2 | 40.4 | 53.0 | 88.8 | 68.1 | 56.3 | 27.4 | 54.7 | 61.8 | 61.6 | 51.2 | 60.4 | 53.5 | 57.7 |
|  | MPT | 59.4 | 82.0 | 86.2 | 56.5 | 89.1 | 68.1 | 62.6 | 34.8 | 67.3 | 62.2 | 62.2 | 52.9 | 67.3 | 73.6 | 63.6 |
|  | DEPT | $44.0_{1.1}$ | $77.4_{6.7}$ | $85.8_{4.4}$ | $59.3_{3.1}$ | $84.1_{2.7}$ | $73.5_{2.8}$ | $63.5_{2.8}$ | $29.3_{2.3}$ | 64.6 | $62.3_{1.3}$ | $62.7_{5.4}$ | $57.5_{1.1}$ | $67.9_{0.9}$ | $75.0_{5.1}$ | 65.1 |
| 16 | PT | 41.5 | 62.3 | 59.9 | 50.9 | 87.8 | 68.1 | 54.7 | 28.5 | 56.7 | 60.3 | 61.9 | 48.9 | 44.2 | 63.5 | 55.8 |
|  | MPT | 61.6 | 84.7 | 90.6 | 63.2 | 89.1 | 70.1 | 64.8 | 32.1 | 69.5 | 64.5 | 63.3 | 49.8 | 67.3 | 78.6 | 64.7 |
|  | DEPT | $61.8_{2.5}$ | $80.3_{1.3}$ | $91.2_{0.5}$ | $77.6_{6.3}$ | $87.1_{1.7}$ | $78.1_{2.3}$ | $71.9_{1.0}$ | $27.1_{1.7}$ | 71.9 | $60.6_{2.8}$ | $66.9_{4.4}$ | $59.6_{0.7}$ | $57.7_{2.7}$ | $78.6_{4.3}$ | 64.7 |
| 32 | PT | 37.0 | 62.3 | 56.7 | 50.9 | 87.5 | 68.1 | 54.7 | 23.2 | 55.1 | 59.2 | 61.7 | 52.6 | 67.3 | 67.8 | 61.7 |
|  | MPT | 63.6 | 88.5 | 91.0 | 75.9 | 89.7 | 74.5 | 59.7 | 30.8 | 71.7 | 63.3 | 68.9 | 53.9 | 67.3 | 82.1 | 67.1 |
|  | DEPT | $63.3_{3.5}$ | $80.1_{0.7}$ | $91.3_{0.5}$ | $80.4_{8.7}$ | $89.2_{0.1}$ | $81.4_{3.3}$ | $72.7_{2.9}$ | $28.6_{2.1}$ | 73.4 | $60.1_{2.7}$ | $67.2_{3.4}$ | $58.0_{0.7}$ | $63.1_{3.6}$ | $82.1_{2.3}$ | 66.4 |

HyperFormer (HF), $(IA)^3$, ATTEMPT (ATP), and MPT on BoolQ, CB, and SciTail datasets. Table 5 presents the performance of DEPT against the vanilla PT and MPT on the GLUE and SuperGLUE benchmark. Experimental results show that vanilla PT struggles with few-shot tasks, indicating the importance of PETL for the PT in few-shot learning tasks as suggested in previous works (Vu et al., 2022; Su et al., 2022). Nevertheless, the performance of DEPT largely benefits from the PETL framework (see Figure 2a). For example, while the vanilla PT obtains an accuracy of 53.5% on SuperGLUE CB dataset and 57.7% on the SciTail dataset when $k$=4, DEPT with PETL achieves an accuracy of 75.0% on SuperGLUE CB dataset and 78.1% on the SciTail dataset, for the same $k$ value. This result supports our first observation about the compatibility of DEPT and PETL approaches. Furthermore, DEPT with transfer learning achieves comparable performance with the variant of the PT, MPT across 14 NLP tasks. Notably, DEPT surpasses the performance of all other variants of the PT (*i.e.,* SPoT, ATTEMPT) and other PEFT approaches, demonstrating our method's efficacy and endorsing our second observation.

**The importance of different learning rates.** Figure 5 presents the experimental results from 3 different learning rate settings to train the soft prompt and the pair of low-rank matrices as follows: (1) use a singular learning rate of 3e-1; (2) use a singular learning rate of 5e-4; (3) apply mixed learning rates (with grid search), where the soft prompt is trained with a larger rate and the pair of low-rank matrices is trained with a lower rate. In our experiments, the first option obtains an average performance

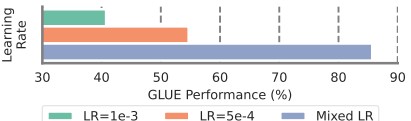

Figure 5: Test results on GLUE benchmark using T5-BASE, showing the importance of training DEPT with different learning rates.

of 40.8 on the GLUE benchmark. The second option exhibits an average performance of 54.7, while the third option demonstrates a largely improved average performance of 85.7 on the GLUE benchmark. This indicates the importance of training DEPT with two different learning rates.

## 4 RELATED WORKS

**Parameter-efficient Fine-tuning.** In contrast to standard fine-tuning and prompt-based fine-tuning (Devlin et al., 2019; Schick & Schütze, 2021; Shi & Lipani, 2023) where full parameters are

updated, parameter-efficient fine-tuning (PEFT) approaches have demonstrated remarkable performance across a wide range of tasks (Wang et al., 2018; Shi et al., 2022; Wu et al., 2023a; Hendriksen et al., 2022; Wu et al., 2023b; Yang et al., 2023) while updating only a limited number of parameters. Adapters (Houlsby et al., 2019), along with its variants, HyperFormer (Karimi Mahabadi et al., 2021) and Compacter (Mahabadi et al., 2021), add new trainable modules (adapters) to each transformer block of the T5 model (Raffel et al., 2020). BitFit (Ben Zaken et al., 2022) limits updates only to the bias parameters, while this method tends to underperform on larger networks (Lialin et al., 2023). Prefix-tuning (Li & Liang, 2021) adds a soft prompt, parameterized by a feed-forward network, to the model input. Diff pruning (Guo et al., 2021) learns a sparse update of a neural network's weights at the cost of more memory usage. FishMask (Sung et al., 2021) also performs sparse updates, but it is computationally intensive and inefficient on contemporary deep learning hardware (Lialin et al., 2023). LoRA (Hu et al., 2021; Yang et al., 2024) employs a straightforward low-rank matrix decomposition to parameterise the weight update. (IA)$^3$ (Liu et al., 2022) scales activations by learned vectors for few-shot learning. LST (Sung et al., 2022b) operates a small transformer network on the side of the pre-trained network, aiming to decrease the training memory. Prompt Tuning (PT) (Lester et al., 2021) appends a trainable soft prompt to the model input embeddings. In comparison to the above-mentioned PEFT approaches, PT uses fewer trainable parameters, which do not proliferate as the model size expands. Mao et al. (2022) introduces a method that combines Prefix-tuning, Adapters, and LoRA through a gating mechanism. DEPT is also applicable to this method and can be easily integrated with other PEFT approaches.

**Transfer Learning for PT.** Recent works aim to enhance the performance of PT through PETL. PPT (Gu et al., 2022) strives to improve the performance of PT (Lester et al., 2021) by further pre-training (Gururangan et al., 2020; Shi et al., 2023), which necessitates a set of hand-crafted, task-specific designs and considerable computational cost. Su et al. (2022) improves PT via prompt transfer across different tasks and models. SPoT (Vu et al., 2022) adopts a single prompt, chosen based on a similarity measure at the cost of a massive search. ATTEMPT (Asai et al., 2022) employs an attention mechanism over the source prompts to initialize the prompt for target tasks at the cost of extra parameters. MPT (Wang et al., 2023b) applies a shared soft prompt across different tasks, while its effectiveness for a broad range of source tasks remains untested. We find that PETL for PT (Asai et al., 2022; Wang et al., 2023b) can efficiently accelerate training convergence, and that PETL for PT is more useful for improving the model performance in the few-shot learning setting for PT (Gu et al., 2022; Wu et al., 2022). However, when extensive labelled datasets are available, training PT or DEPT for additional steps typically leads to performance improvements.

## 5 EPILOGUE

**Conclusion.** In this work, we propose Decomposed Prompt Tuning (DEPT), which substantially improves the efficiency of the vanilla PT in terms of time and memory while delivering competitive or even superior performance compared to the state-of-the-art PEFT methods. Remarkably, DEPT efficiency amplifies with increasing model sizes, making it exceptionally apt for LLMs. Our further analysis shows the compatibility of DEPT with PETL approaches and highlights its versatility across diverse model architectures and scales.

**Limitations and Future Work.** We outline several limitations in our work: (1) the main limitation of DEPT is the introduction of extra hyperparameters for tuning, *e.g.,* the learning rate of the low-rank matrices. This might introduce some additional computational overhead during the hyperparameter optimization phase of model training. In our work, we train DEPT up to 300k steps (in a data-rich setting) following (Vu et al., 2022) with a careful search for optimal learning rates, which may increase training costs. However, the number of training steps might be efficiently reduced by PETL, which we plan to investigate in future work. In addition, it is important to note that the model training process is a one-time event, while model inference is not. In this context, the efficiency benefits of DEPT become especially valuable; (2) the number of trainable parameters in DEPT depends on the maximum sequence length $s$. In this work, we have limited our evaluation to tasks with hundreds of input tokens. Future work could explore the performance of DEPT when $s$ is extremely large; and (3) our research focuses on leveraging prompting techniques for LMs, where previous studies (Bender & Koller, 2020; Brown et al., 2020; Bender et al., 2021) have already addressed concerns and potential hazards linked to LMs.

ACKNOWLEDGMENTS

The authors express their gratitude to the ICLR reviewers and area chairs for their insightful discussions. Zhengxiang is funded by the Research Studentship from University College London (UCL).

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

## APPENDIX OVERVIEW

The appendix is structured as follows:

**Appendix §A** provides a visualization of the model performance against the number of trainable parameters on the GLUE and SuperGLUE benchmarks.

**Appendix §B** presents the additional experimental results, including using a larger size of language models (LLAMA-2 and TB-3B) and testing the impact of different lengths of soft prompts.

**Appendix §C** provides a brief description of all datasets used in this work.

**Appendix §D** provides implementation details and hyperparameters for all comparison methods used in our experiments.

**Appendix §E** provides further discussion regarding intuitions and related works.

## A  MODEL PERFORMANCE AGAINST THE PARAMETER-EFFICIENCY

We visualize the experimental results in Table 1, as shown in Figure 6. The visualization shows that our proposed method DEPT outperforms other PEFT approaches and full fine-tuning baselines on the GLUE and SuperGLUE benchmark (y-axis) while updating only a small number of trainable parameters (x-axis).

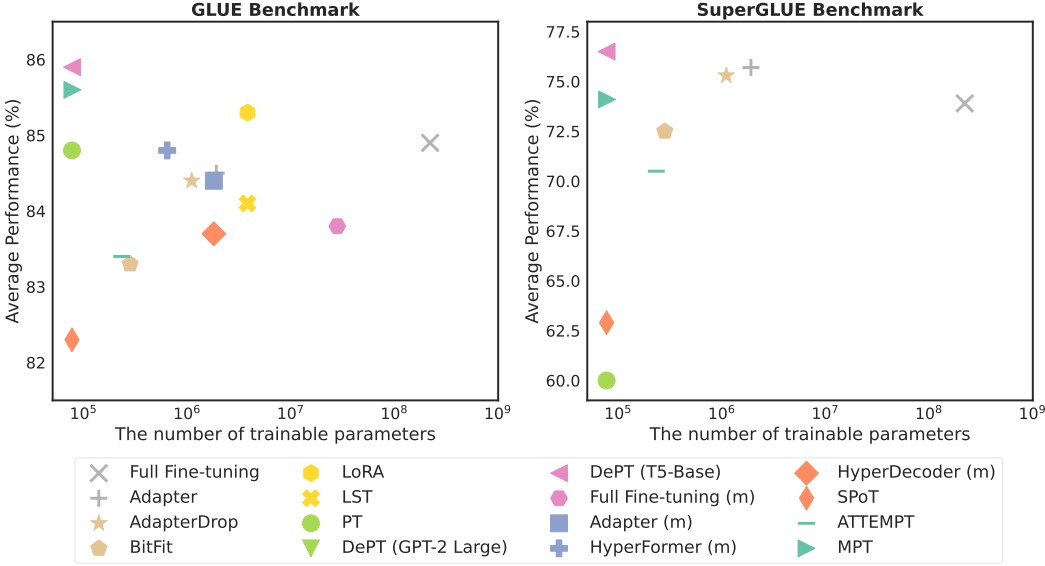

Figure 6: The average performance against the number of trainable parameters on the GLUE and SuperGLUE benchmark using the T5-BASE model.

## B  ADDITIONAL EXPERIMENTS

**LLAMA-2.** We evaluate the performance and inference speed of our proposed method DEPT using LLAMA-2-7B and LLAMA-2-13B (Touvron et al., 2023) on the SST-2 dataset. In our experiment, the soft prompt length for the vanilla PT is set to $l = 100$. For DEPT, we set the soft prompt length to $m = 60$ and select a rank of $r = 40$ for the low-rank matrices. As shown in Table 6, our experimental results suggest that DEPT not only outperforms the vanilla PT in terms of test accuracy but also improves the speed of inference. We only limit our evaluation of DEPT to the

SST-2 dataset due to the high computational expenses. We will do our best to get the necessary resources to further probe the performance of DEPT, aiming to deliver a more exhaustive evaluation in future work.

| Method | Prompt Tuning | | DEPT (ours) | |
|---|---|---|---|---|
| | Test Acc | Inference samples per second | Test Acc | Inference samples per second |
| LLAMA-2-7B | 94.48 | 3.895 | 94.95 | 4.857 |
| LLAMA-2-13B | 95.99 | 2.083 | 96.01 | 2.835 |

Table 6: Test results using LLAMA-2-7B and LLAMA-2-13B on the SST-2 dataset.

**T5-3B.** We evaluate the performance and inference speed of our proposed method DEPT using the T5-3B model. We report the average performance on the Glue dataset as well as inference speed, measured in inference samples per second. As shown in Table 7, our findings indicate that DePT (m=60, r=30) outperforms PT in terms of inference speed by 37%. This suggests the advantage of DePT increases as the model size increases.

| Method | Average Glue Performance | Inference samples per second |
|---|---|---|
| DEPT (m=60, r=30) | 86.4 | 8.9 |
| PT (m=100) | 85.6 | 6.5 |

Table 7: Test results using T5-3B on the Glue Benchmark.

**Different prompt lengths.** We have performed additional experiments regarding different prompt lengths, as shown in the Table below. Specifically, we have increased the size of trainable parameters in both DEPT and PT by a factor of two. We use the T5-BASE as the backbone. As shown in Table 8, we report the average performance on the Glue dataset as well as inference speed, measured in inference samples per second. Our findings indicate that DEPT (m=120, r=60) outperforms PT in terms of inference speed by 34%. We believe that this performance advantage can be further enhanced by reducing the value of $m$, which represents the length of the soft prompt. To provide a concrete example, on the SST-2 dataset, DEPT can achieve an inference speed of 77.2 samples per second, while PT can only infer 57.4 samples per second. This suggests the advantage of DEPT over PT increases as the model size increases.

| Method | Average Glue Performance | Inference samples per second |
|---|---|---|
| DEPT (m=120, r=60) | 86.0 | 54.8 |
| PT (m=200) | 85.2 | 40.8 |

Table 8: The impact of using longer soft prompt length. Test results using T5-BASE on the Glue Benchmark.

## C  DATASET

In this work, we use 23 popular datasets from previous few-shot learning and PEFT research. We limit the maximum training data number of Yelp-2 to 100k samples. We train MNLI with longer steps, 200k steps in total. For the GLUE dataset, we use the HuggingFace dataset[2]. For the Super GLUE dataset, we use the HuggingFace dataset[3]. For MRQA 2019 Shared Task and other datasets, we use the HuggingFace dataset[4].

| GLUE Benchmark | | | | | | |
|---|---|---|---|---|---|---|
| **Dataset** | **Source** | **Target** | **#Train** | **#Valid** | **#Test** | **Type** |
| MNLI | 31.8 | 1.0 | 392,702 | 9,832 | 9,815 | NLI |
| QQP | 24.1 | 1.0 | 362,846 | 1,000 | 40,431 | Paraphrase |
| QNLI | 38.4 | 1.0 | 103,743 | 1,000 | 5,463 | NLI |
| SST-2 | 10.4 | 1.0 | 66,349 | 1,000 | 872 | Sentiment |
| STS-B | 21.9 | 1.0 | 5,749 | 750 | 750 | Sent. Similarity |
| MRPC | 45.9 | 1.0 | 3,668 | 204 | 204 | Paraphrase |
| RTE | 54.4 | 1.0 | 2,490 | 138 | 139 | NLI |
| CoLA | 8.7 | 1.0 | 8,551 | 521 | 522 | Acceptability |
| SuperGLUE Benchmark | | | | | | |
| **Dataset** | **Source** | **Target** | **#Train** | **#Valid** | **#Test** | **Type** |
| MultiRC | 286.1 | 1.0 | 27,243 | 2,424 | 2,424 | Question Answering |
| BoolQ | 108.3 | 1.0 | 9,427 | 1,635 | 1,635 | Question Answering |
| WiC | 18.4 | 1.0 | 5,428 | 319 | 319 | Word Sense Disambiguation |
| WSC | 28.1 | 1.0 | 554 | 52 | 52 | Common Sense Reasoning |
| CB | 64.6 | 1.0 | 250 | 28 | 28 | NLI |
| ReCoRD | 210.7 | 1.5 | 137,484 | 1,370 | 15,176 | Common Sense Reasoning |
| MRQA 2019 Shared Task | | | | | | |
| **Dataset** | **Source** | **Target** | **#Train** | **#Valid** | **#Test** | **Type** |
| NaturalQuestions | 242.7 | 4.5 | 103,071 | 1,000 | 12836 | Question Answering |
| HotpotQA | 225.7 | 2.6 | 71,928 | 1,000 | 5,901 | Question Answering |
| SearchQA | 942.8 | 2.0 | 116,384 | 1,000 | 16,980 | Question Answering |
| NewsQA | 615.5 | 5.1 | 73,160 | 1,000 | 4,212 | Question Answering |
| Other Datasets | | | | | | |
| **Dataset** | Source | Target | **#Train** | **#Valid** | **#Test** | **Type** |
| WinoGrande | 23.8 | 1.0 | 39,398 | 1,000 | 1,267 | Common Sense Reasoning |
| YelpPolarity | 134.0 | 1.0 | 100,000 | 1,000 | 38,000 | Sentiment |
| SciTail | 30.8 | 1.0 | 23,596 | 652 | 652 | NLI |
| PAWS | 44.7 | 1.0 | 4,9401 | 8,000 | 8,000 | Sent. Similarity |
| Vision Language Tasks (#Images & #Texts) | | | | | | |
| Visual Question Answering | - | - | 113.2K/605.1K | 5.0K/26.7K | 5.0K/26.3K | Question Answering |
| MS CoCo Caption | - | - | 113.2K/566.8K | 5.0K/5.0K | 5.0K/5.0K | Caption Generation |

Table 9: The datasets evaluated in this work. Source indicates the average length of the source sentences in the training set. Target indicates the average length of the target sentences in the training set. STS-B is a real-valued regression task over the interval $[0, 5)$. Note that we only sample examples from the original training set in our few-shot experiments.

| **Hyperparameter** | **Assignment** |
|---|---|
| number of steps | 30,000 steps (evaluate every 1,000 steps) |
| batch size | 16 |
| maximum learning rate ($\alpha_1$) | 3e-1, 4e-1, 5e-1 |
| maximum learning rate ($\alpha_2$) | 1e-04, 5e-4, 1e-03 |
| length of the soft prompt ($m$) | 20, 40, 60, 80 |
| maximum sequence length | 256 |
| learning rate optimizer | AdamW |
| Adam epsilon | 1e-6 |
| Adam beta weights | 0.9, 0.98 |
| learning rate scheduler | Warmup linear |
| Weight decay | 0.01 |
| Warmup proportion | 0.06 |

Table 10: Hyperparameters for Prompt Tuning and DEPT.

## D   IMPLEMENTATION DETAILS

Our code is implemented using Pytorch[5], Huggingface Transformers[6], and Huggingface PEFT[7]. Below, we provide a comprehensive list of the hyperparameters used in our code. In our work, we mainly cite the experimental results from the previous works Asai et al. (2022); Wang et al. (2023b); Sung et al. (2022b). In addition, we train LoRA with up to 200k steps. We search the learning rate within the set {5e-4, 1e-4, 5e-5, 1e-5}. We set the rank as 35. We choose a batch size of 32. We find that training LoRA on the MRQA dataset presents challenges, despite conducting a thorough search for optimal learning rates and training steps. The reasons for these difficulties remain uncertain. For prompt tuning and DEPT, as shown in Table 10, we conduct a grid search for learning rates. For the soft prompt, we search the learning rate within the set {3e-1, 4e-1, 5e-1}. For the low-rank matrice pairs, we search the learning rate within the set {1e-04, 5e-4, 1e-03, 5e-03}. We choose a batch size of 16. We typically use the max sequence length as 256 except for the SuperGLUE-MultiRC, where the max sequence length is 348. In each trial, we train the model for 30,000 steps, evaluate performance every 1,000 steps, and select the best checkpoint based on optimal performance on the evaluation set. For the large dataset with more than 100,000 training examples, we follow the prior work (Vu et al., 2022) to train the vanilla PT and our proposed method DEPT with up to 300,000 steps. Training more steps helps improve the performance of the vanilla PT for the large dataset. The best performance is determined by the relevant evaluation metric. We train the T5 model from the original checkpoint rather than the LM-adapted 1.1 version (Lester et al., 2021).

## E   FURTHER DISCUSSION

**Intuition.**   The intuition of DEPT is that (1) given the same number of trainable parameters, allowing some updates for word embeddings will improve the performance; and (2) a shorter soft prompt will improve the efficiency. To illustrate, the previous study (Wingate et al., 2022) has shown that a soft prompt can interpolate between many token embeddings, enabling the representation of more abstract concepts compared to relying on a single discrete token. However, the soft prompt in the PT is consistently added at the beginning of the frozen word embedding. In contrast, we propose DEPT, which decomposes the long soft prompt into a short soft prompt and a pair of low-rank matrices. This approach can (1) reduce the length of the soft prompt for better efficiency; and (2) permit representation updates within the frozen word embedding, thereby increasing the adaptability of input representations that were previously unavailable.

**Related works with similar titles.**   The meaning of "compose" and the method are fundamentally different between previous works (Khot et al., 2022; Nayak et al., 2022) and our work. Specifically, Decomposed Promptin (Khot et al., 2022) focuses on in-context learning, without the need to update parameters. Decomposed Prompting aligns closely with the work of chain-of-thoughts and self-consistency. In addition, CSP (Nayak et al., 2022) treats the attributes and objects that are composed to define classes as learnable tokens within the vocabulary. In contrast, our proposed method DePT does not train soft prompts associated with any vocabulary token, nor does it add additional tokens to the vocabulary. The main goal of DePT is to improve the efficiency of Prompt Tuning (PT) due to the increased input sequence issue.

**Comparison between Prompt Tuning (PT) and LoRA.**   We would like to discuss the comparison between Prompt Tuning (PT) and LoRA, as our work aims to improve the PT, in the following points:

- **Relative Performance of LoRA and PT.** When adapting language models (LMs) to specialised domains, like mathematical reasoning, which requires much different knowledge than what LLMs have been trained on, LoRA may perform better than Prompt Tuning (PT). However, in case tasks have already been somewhat understood by LMs and the key challenge is just to properly prompt the LMs, PT can be the better option. PT modifies minimal

---

[2]https://huggingface.co/datasets/glue
[3]https://huggingface.co/datasets/super_glue
[4]https://huggingface.co/lucadiliello
[5]https://pytorch.org/
[6]https://github.com/huggingface/transformers
[7]https://github.com/huggingface/peft

model parameters, focusing instead on improving the input prompt, which has been proven more effective than LoRA in prior studies (Asai et al., 2022; Wang et al., 2023b).

- **Specific Use Cases for PT.** PT offers advantages in particular cases. For example, soft prompts can be used to compress few-shot examples in the prompt or long context (Chevalier et al., 2023; Wingate et al., 2022). While the number of trainable parameters is low, LoRA updates the weight matrices across the whole model. In contrast, PT only improves the input of the LM through the soft prompt, which helps the model focus on understanding the task and context better rather than learning new knowledge.

- **Parameter Efficiency.** Unlike LoRA, which requires trainable parameters at each layer, PT's trainable parameters are more concentrated and less extensive.

- **Parameter-efficient transfer learning (PEFT) Framework.** Framework. PETL framework (e.g., Asai et al. (2022); Wang et al. (2023b)) can effectively improve the performance of the PT and make it easier to use. In our work, we have demonstrated that our approach is compatible with the PEFT framework.

