# OpenReview forum: "DePT: Decomposed Prompt Tuning for Parameter-Efficient Fine-tuning"
_ICLR.cc/2024/Conference — ICLR 2024 poster_

### Official Review · Reviewer_XBEs · 2023-10-13

**Soundness:** 1 poor
**Presentation:** 2 fair
**Contribution:** 2 fair
**Rating:** 6
**Confidence:** 5

**Summary:**

The paper proposes Decomposed Prompt Tuning (DePT), which decomposes the soft prompt into a shorter soft prompt and a pair of low-rank matrices optimized with two different learning rates. The authors conduct experiments to evaluate the effectiveness and efficiency of the proposed DePT.

**Strengths:**

- the paper is easy to follow
- extensive experiments on simple datasets are conducted to evaluate the proposed method

**Weaknesses:**

- **motivation is weak**:
  - soft prompts are already very parameter-efficient; it is not necessary to reduce the #parameters in prompt tuning
  - usually, the length of soft prompts is small, e.g., 8, 16, 32; the additional inference cost is minor compared with the current LLM, which can accept thousands of tokens as input
  - though prompt tuning is sensitive to initialization, there are some recent methods to deal with this problem, e.g.,
    - Effective Structured Prompting by Meta-Learning and Representative Verbalizer, ICML 2023
    - MetaPrompting: Learning to learn better prompts, COLING 2022
- for the method, the idea of using a LoRA matrix to approximate part of soft prompts is odd to me. Can the authors visualize the learned A and B matrices to explain why DePT is better than Prompt tuning?
- the tasks used in experiments are too simple; better to try more challenging tasks, like GeoQuery (Zelle & Mooney, 1996), NL2Bash (Lin et al., 2018), WebQS (Berant et al., 2013).
- datasets in GLUE and SuperGLUE are sensitive to hyperparameters. What are the hyperparameters for each dataset?
-  In Table 3, DePT is much worse than FT on VQA, but can perform better on MSCOCO; why?
- how to initialize the soft prompt in prompt tuning? Usually, they can be initialized randomly or use the embeddings of label tokens.

**Questions:**

See the questions mentioned above.

---

> ### Author Response · Authors · 2023-11-16
> **Author Rebuttal by Authors (1/2)**
>
> We appreciate the effort and time by the reviewer (XBEs).
>
> `soft prompts are already very parameter-efficient; it is not necessary to reduce the #parameters in prompt tuning`
>
> We thank the reviewer for the feedback. We agree that prompt tuning is very parameter-efficient. However, this weakness is not applicable to our work, because we do **NOT** reduce the number of parameters compared to prompt tuning.
>
> `Usually, the length of soft prompts is small, e.g., 8, 16, 32; the additional inference cost is minor compared with the current LLM, which can accept thousands of tokens as input`
>
> We would like to express our appreciation for the reviewer's feedback. However, we respectfully present a different perspective:
>
> - Previous works [1,2,3,4,5,6,7,8] typically use soft prompts with a default length exceeding 100 virtual tokens. To support this point, we can directly quote previous work [5], which states that "***for hard sequence tasks, usually, a longer prompt than 100 would be helpful***". Similar claims can be found throughout the literature. To the best of our knowledge, there is no existing work that utilises soft prompts with a length of fewer than 32 virtual tokens while accepting thousands of tokens as the input.
> - Let's take a step back and consider the situation more broadly. Assuming that we can only add at most 32 virtual tokens to the soft prompt, our proposed method allows us to achieve a highly competitive or even better performance than using around 100 virtual tokens, with minor operations and costs (*i.e.,* add the multiplication of low-rank matrices element-wisely to frozen word embeddings at the input sequence). We believe that this should be regarded as a notable strength of our work.
>
> `though prompt tuning is sensitive to initialization, there are some recent methods to deal with this problem e.g., Effective Structured Prompting by Meta-Learning and Representative Verbalizer, ICML 2023. MetaPrompting: Learning to learn better prompts, COLING 2022.`
>
> We thank the reviewer for the feedback. We respectively argue that this weakness is not applicable to our work because:
> - Our main goal is **NOT** to address the initialisation problem of the prompt tuning. Instead, our work primarily addresses the increased computational demands, in terms of training, inference time, and memory costs, which arise from the increased input sequence length due to adding the soft prompt.
> - In our work, we have extensively discussed these Parameter-efficient Transfer Learning (PETL) approaches by previous works, which aim to solve the initialisation problem. In Section 3.4 of our paper, we have empirically demonstrated that our proposed method complements PETL approaches in the few-shot learning setting. This substantiates our position that the initialisation challenge, while relevant in other contexts, is not applicable to our method.
>
> We appreciate the reviewer for suggestions on related works. We will make sure to discuss these works in our revised version.
>
> `the tasks used in experiments are too simple; better to try more challenging tasks, like GeoQuery (Zelle & Mooney, 1996), NL2Bash (Lin et al., 2018), WebQS (Berant et al., 2013).`
>
> We thank the reviewer for this suggestion. However, we respectfully disagree with the reviewer:
>
> - All other reviewers (`mz3i`,`BZux`,`yLsP`) have commented positively on the thoroughness of our experiments, suggesting that they find our current evaluation to be adequate.
> - Our work has already included a comprehensive evaluation of 23 datasets. These tasks range from natural language inference and sentiment analysis to open-domain question answering, common-sense reasoning, and even visual-language tasks. We believe that these tasks collectively represent a diverse and challenging set, and we do not consider them to be simple.
> - Our experiments adhere to the settings and tasks used in previous works [1,2,3,4,5,6,7,8], strictly following the two most recent works [7,8].
> - WebQs is a question-answering task, while GeoQuery and NL2Bash are generation tasks. Both of these task types have already been covered by the existing datasets in our work.
>
> ### References:
>
> [1] The Power of Scale for Parameter-Efficient Prompt Tuning. EMNLP 2021.
>
> [2] On Transferability of Prompt Tuning for Natural Language Processing. NAACL 2022.
>
> [3] SPoT: Better Frozen Model Adaptation through Soft Prompt Transfer. ACL 2022.
>
> [4] PPT: Pre-trained Prompt Tuning for Few-shot Learning. ACL 2022.
>
> [5] P-Tuning: Prompt Tuning Can Be Comparable to Fine-tuning Across Scales and Tasks. ACL 2022.
>
> [6] PSP: Pre-trained Soft Prompts for Few-Shot Abstractive Summarization. COLING 2022.
>
> [7] ATTEMPT: Parameter-Efficient Multi-task Tuning via Attentional Mixtures of Soft Prompts. EMNLP 2022.
>
> [8] Multitask Prompt Tuning Enables Parameter-Efficient Transfer Learning. ICLR 2023.

---

> > ### Author Response · Authors · 2023-11-16
> > **Author Rebuttal by Authors (2/2)**
> >
> > `datasets in GLUE and SuperGLUE are sensitive to hyperparameters. What are the hyperparameters for each dataset`
> >
> > We appreciate the reviewer's question. For the soft prompt, we search the learning rate within the set {3e-1, 4e-1, 5e-1}. For the low-rank matrice pairs, we search the learning rate within the set {1e-04, 5e-4, 1e-03}. For more details, please refer to our Section D: Implementation details in the Appendix and our anonymous code repository (https://anonymous.4open.science/r/DePT-8F43/README.md).
> >
> > `for the method, the idea of using a LoRA matrix to approximate part of soft prompts is odd to me. Can the authors visualize the learned A and B matrices to explain why DePT is better than Prompt tuning?`
> >
> > We thank the reviewer for this suggestion. The intuition of our method is that (1) given the same number of trainable parameters, allowing some updates for word embeddings will improve the performance; and (2) shorter soft prompt will improve the efficiency. To illustrate, the previous study [9] has shown that a soft prompt can interpolate between many token embeddings, enabling the representation of more abstract concepts compared to relying on a single discrete token. However, the soft prompt in the Prompt Tuning is consistently added at the beginning of the frozen word embedding. In contrast, we propose DePT, which decomposes the long soft prompt into a short soft prompt and a pair of low-rank matrices. This approach can (1) reduce the length of the soft prompt for better efficiency; and (2) permit representation updates within the frozen word embedding, thereby increasing the adaptability of input representations that were previously unavailable.
> >
> > `In Table 3, DePT is much worse than FT on VQA, but can perform better on MSCOCO; why?`
> >
> > We appreciate the reviewer's observation regarding the performance discrepancy between DePT and FT on VQA in Table 3. Our intention is to illustrate that DePT may not perform well in certain tasks. As discussed in Section 3.2 of our paper, this variation suggests that in specific tasks, the use of a greater number of trainable parameters could be beneficial. In essence, our work aims to show the versatility of our proposed method across a range of tasks while acknowledging that the effectiveness of PEFT approaches can be task-dependent. We believe that this insight contributes to a better understanding of PEFT approaches.
> >
> > `how to initialize the soft prompt in prompt tuning? Usually, they can be initialized randomly or use the embeddings of label tokens.`
> >
> > We appreciate the reviewer's question. For the task with a large number of training examples (Table 1 and 2), we initiate the soft prompt using word embedding from the vocabulary. In the few-shot learning experiments (Table 4 and 5), we initiate the soft prompts from the pre-trained soft prompts on one of the source tasks (MNLI, QQP, SST-2, SQUAD, and ReCoRD), as we mentioned in the Implementation details of Section 3.1 and Appendix D.
> >
> > ### References:
> >
> > [9] Prompt Compression and Contrastive Conditioning for Controllability and Toxicity Reduction in Language Models. Findings of EMNLP 2022.

---

> > > ### Comment · Reviewer_XBEs · 2023-11-21
> > > **Thanks for your response**
> > >
> > > Thank for the author's reply, which resolved my major concerns.
> > > Thus, I increased my score to 6.

---

> > > > ### Author Response · Authors · 2023-11-21
> > > > **Official Comment by Authors**
> > > >
> > > > We are very glad to hear that our response is helpful! Thank you so much for taking the time to revise your review!

---

### Official Review · Reviewer_mz3i · 2023-10-30

**Soundness:** 4 excellent
**Presentation:** 4 excellent
**Contribution:** 2 fair
**Rating:** 6
**Confidence:** 4

**Summary:**

The work introduces decomposed prompt tuning for parameter-efficient fine-tuning, or DEPT. They propose to reduce the prompt and add a decomposable matrix to the word embeddings. The experiments on GLUE and SuperGLUE show the method's effectiveness while being more efficient than prompt tuning. The few-shot learning experiments show that the method is competitive with other parameter-efficient fine-tuning methods with more parameters. Overall, the method improves prompt tuning while being 20% efficient in training.

**Strengths:**

**Paper quality.** The paper is well written. The organization of the paper is clear and well thought out. I enjoyed reading the paper.

**Extensive experiments.** The paper extensively experiments with improved results compared to prompt tuning while being more efficient during training and inference. The authors have done a great job comparing the work with other recent methods in the literature and show that DEPT outperforms them on GLUE and SuperGLUE. They further provide evidence of their efficiency on GPT and Llama2 models.

**Code.** The code is well organized and easy to understand (see https://anonymous.4open.science/r/DePT-8F43/README.md).

**Weaknesses:**

**The architecture is not well motivated.**
The architecture appears to be a combination of prompt tuning and LoRA. But, unlike LoRA, DEPT still suffers from prompt length compared to architectures at inference time. While DEPT can also achieve the same inference speed as the base model, like LoRA, when the prompt length is 0, in Figure 3, we see that the performance is about 20 points below the DEPT performance reported in Table 1. Furthermore, decomposing the prompts does not offer any conceptual understanding or insight into the learned prompts.

**Experiments with only smaller models (<1B model parameters).**
All experiments in the main paper are conducted on smaller models which hurt the technical contribution of the work. The paper presents improved results over several parameter-efficient methods, by a small margin, on models that have less than 1 billion parameters. The authors have included results of Llama2 in Appendix B but show improvements over prompt tuning on only one dataset (SST-2). It would be of interest to the community to know if the work is applicable to larger models say T5-XL, T5-XXL, or Llama2 on more datasets.

**Missing experiments.**
I noticed that the authors do not include results on SuperGLUE for LoRA in Table 1. Since the results for DEPT and LoRA are so close on GLUE, I would be curious to see how close the results are on SuperGLUE.

**Suggestions.**
- Related work: The title of your work is similar to the below papers [a,b] but the proposed work differs. You could choose to cite them and clarify the differences in the next version of the paper.
- Move the 3.3 #2 (DEPT grows more efficient with model size increases) to the Appendix and move Llama2 to the main paper as it is obvious that the inference time will improve with model parameter size. The Llama2 results, on the other hand, are more interesting.
- For completeness, it would be great if the authors could include the GPT-2 results in the Appendix on GLUE and SuperGLUE. I’d be curious to see if there are any changes in the performance.

References:

[a] Decomposed prompting: A modular approach for solving complex tasks. ICLR 2023.

[b] ​​Learning to compose soft prompts for compositional zero-shot learning. ICLR 2023.

**Questions:**

- In implementation details, you have included that the soft prompts and low rank matrices are initialized from soft prompts derived from one of the source tasks. Could you clarify this detail? It is unclear why this is necessary.
- Do you think $\mathbf{BA}$ could use a scaling factor like $\frac{\alpha}{r}$? It is possible that this could be causing the low performance due to when only the decomposed matrices are trained.
- In the background, you have mentioned that $s$ is the maximum sequence length. How would you set $s$ for Llama2 or models that do not have a fixed maximum sequence length?

---

> ### Author Response · Authors · 2023-11-16
> **Author Rebuttal by Authors (1/2)**
>
> We appreciate the effort and time by the reviewer (mz3i).
>
> `The architecture is not well motivated. The architecture appears to be a combination of prompt tuning and LoRA. But, unlike LoRA, DEPT still suffers from prompt length compared to architectures at inference time.`
>
> We thank the reviewer for the insightful suggestion. In response, we would like to discuss the comparison between Prompt Tuning (PT) and LoRA, as our work aims to improve the PT, in the following points:
>
> - **Relative Performance of LoRA and PT.** When adapting language models (LMs) to specialised domains, like mathematical reasoning, which requires much different knowledge than what LLMs have been trained on, LoRA may perform better than Prompt Tuning (PT). However, in case tasks have already been somewhat understood by LMs and the key challenge is just to properly prompt the LMs, PT can be the better option. PT modifies minimal model parameters, focusing instead on improving the input prompt, which has been proven more effective than LoRA in prior studies [1,2].
> - **Specific Use Cases for PT.** PT offers advantages in particular cases. For example, soft prompts can be used to compress few-shot examples in the prompt or long context [3,4]. While the number of trainable parameters is low, LoRA updates the weight matrices across the whole model. In contrast, PT only improves the input of the LM through the soft prompt, which helps the model focus on understanding the task and context better rather than learning new knowledge.
> - **Parameter Efficiency.** Unlike LoRA, which requires trainable parameters at each layer, PT's trainable parameters are more concentrated and less extensive.
> - **Parameter-efficient transfer learning (PETL) Framework.**  PETL framework (e.g., [1,2]) can effectively improve the performance of the PT and make it easier to use. In our work, we have demonstrated that our approach is compatible with PEFL framework.
>
> `Experiments with only smaller models (<1B model parameters).`
>
> We respectfully disagree with the reviewer. While we acknowledge the importance of exploring this avenue, we are currently constrained by limited computing resources. This is a common issue faced by many researchers and practitioners. Therefore, the findings of this paper are relevant not just to us but also to others in a similar situation. However, also in response to this complain, we have conducted additional experiments on T5-3B. We report the average performance on the Glue dataset as well as inference speed, measured in inference samples per second. Our findings indicate that DePT (m=60, r=30) outperforms PT in terms of inference speed by 37%. This suggests the advantage of DePT increases as the model size increases.
>
> |  | Average Glue Performance | Inference Speed |
> | --- | --- | --- |
> | DePT (m=60, r=30) | 86.4 | 8.9 |
> | PT (m=100) | 85.6 | 6.5 |
>
> (*m indicates the length of the soft prompt*, *r indicates the rank of low rank matrices*)
>
> We will do our best to get the necessary resources to investigate this important direction, and we leave this to future work.
>
> `Missing experiments. I noticed that the authors do not include results on SuperGLUE for LoRA in Table 1. Since the results for DEPT and LoRA are so close on GLUE, I would be curious to see how close the results are on SuperGLUE.`
>
> We thank the reviewer for the insightful suggestion. In response, we have add LoRA and IA3 as baselines and filled the missing number for Table 1. Specifically, we train LoRA and IA3 with 30k steps with a batch size of 16. We use the rank as 35 for LoRA. We perform a learning rate search among 3e-3, 5e-4, 1e-4, and 5e-5, and report the test results of the best model on the development set. Our experiment show that DePT obtains better mean performance on SuperGlue benchmark.
>
> |                                | Multirc | Bool   | WiC   | WSC | CB    | Mean |
> |----------------    |-------  |-------|----    |----    |------|------|
> |LoRA                       | 72.6     | 81.3    |  68.3 | 67.3   | 89.3  | 75.8 |
> |IA3                           | 73.3    | 80.9    |  67.7 | 73.1    | 85.7  | 76.1  |
>
> ### References:
>
> [1] ATTEMPT: Parameter-Efficient Multi-task Tuning via Attentional Mixtures of Soft Prompts. EMNLP 2022.
>
> [2] Multitask Prompt Tuning Enables Parameter-Efficient Transfer Learning. ICLR 2023.
>
> [3] Prompt Compression and Contrastive Conditioning for Controllability and Toxicity Reduction in Language Models, EMNLP 2022.
>
> [4] Adapting Language Models to Compress Contexts. EMNLP 2023.

---

> > ### Author Response · Authors · 2023-11-16
> > **Author Rebuttal by Authors (2/2)**
> >
> > `Related work: The title of your work is similar to the below papers [a,b] but the proposed work differs. You could choose to cite them and clarify the differences in the next version of the paper.`
> >
> > We thank the reviewer for the insightful suggestion. We appreciate the opportunity to clarify that: the meaning of “compose” and the method are fundamentally different between previous works [5,6] and our work. Specifically, Decomposed Prompting [5] focuses on in-context learning, without the need to update parameters. Decomposed Prompting aligns closely with the work of chain-of-thought and self-consistency. In addition, CSP [6] treats the attributes and objects that are composed to define classes as learnable tokens within the vocabulary. In contrast, our proposed method DePT does not train soft prompts associated with any vocabulary token, nor does it add additional tokens to the vocabulary. The main goal of DePT is to improve the efficiency of Prompt Tuning (PT) due to the increased input sequence issue. We have mentioned these two works in the Appendix of our revised version.
> >
> > `Move the 3.3 #2 (DEPT grows more efficient with model size increases) to the Appendix and move Llama2 to the main paper as it is obvious that the inference time will improve with model parameter size. The Llama2 results, on the other hand, are more interesting.`
> >
> > We thank the reviewer for the suggestion. We believe we can easily address this in the camera-ready version, once we have collected more results using llama-2 models.
> >
> > `For completeness, it would be great if the authors could include the GPT-2 results in the Appendix on GLUE and SuperGLUE. I’d be curious to see if there are any changes in the performance.`
> >
> > We thank the reviewer for the insightful suggestion. In response, we have added this part of the results to Figure 6 in Appendix A.
> >
> > `In implementation details, you have included that the soft prompts and low-rank matrices are initialized from soft prompts derived from one of the source tasks. Could you clarify this detail? It is unclear why this is necessary.`
> >
> > We thank the reviewer for the question. This is due to the limitation of Prompt Tuning, which is hard to initialise soft prompts especially when the number of training data is not large, which is discussed in Section 1 of our paper. To mitigate this issue, previous works (e.g., [1,2]) have proposed Parameter-efficient Transfer Learning (PETL). PETL initially trains soft prompts on source tasks and then fine-tunes them on target tasks. PETL can effectively improve the performance of the PT and make it easier to use. In our work, we have demonstrated that our approach is compatible with PEFL approaches in the few-shot learning experiments.
> >
> > `Do you think BA could use a scaling factor like a/r? It is possible that this could be causing the low performance due to when only the decomposed matrices are trained.`
> >
> > We thank the reviewer for the insightful suggestion. We have explored the possibility of incorporating a scaling factor, such as a/r, within the BA. However, our experiments indicate that this approach did not lead to an improvement in performance.  We have discussed them in the Appendix of our revised version.
> >
> > `In the background, you have mentioned that s is the maximum sequence length. How would you set s for Llama2 or models that do not have a fixed maximum sequence length?`
> >
> > We thank the reviewer for the insightful suggestion. We appreciate the opportunity to clarify: the max sequence length is for truncation and padding, and it does not represent the maximum input length for the models.
> >
> > ### References:
> >
> > [5] Decomposed prompting: A modular approach for solving complex tasks. ICLR 2023.
> >
> > [6] Learning to compose soft prompts for compositional zero-shot learning. ICLR 2023.

---

> > > ### Comment · Reviewer_mz3i · 2023-11-21
> > > **Reply to the Authors**
> > >
> > > Thank you for your response. I am happy to see that DEPT performs better than all the previous parameter-efficient learning methods. I appreciate you clarifying most of my queries and updating your paper. However, the intuition behind the architecture (included in the updated paper) is still not convincing and the overall results are better than previous methods but they are not significantly higher. These factors could limit the impact of the work. For this reason, I am going to keep my score at 6.

---

> ### Author Response · Authors · 2023-11-21
> **Official Comment by Authors**
>
> Thank you so much for your continued discussion and feedback! We appreciate the opportunity to clarify and discuss:
> - **Architecture Justification**. We understand the reviewer’s question about the architecture's intuition. It's important to highlight that the series of Prompt Tuning works, which form the core of our architecture, has shown distinct advantages in certain tasks over other Parameter-Efficient Fine-Tuning (PEFT) approaches, as we discussed in the previous reply. Therefore, we believe that our work, with the goal of improving the design of prompt tuning, remains a promising avenue for future research.
> - **Model Performance and Efficiency**. Our work focuses not only on improving the model performance. An important contribution of our work is addressing the increased computational demands, in terms of training, inference time, and memory costs, which arise from the increased input sequence length due to adding the soft prompt. Although the improvement in performance may not be dramatic, the efficiency gains are notable. For example, as in our previous reply, when T5-3B is used as the backbone, we find the DePT (m=60, r=30) outperforms PT in terms of inference speed by 37%, underscoring the efficiency enhancements our proposed method has achieved.

---

> > ### Comment · Reviewer_mz3i · 2023-11-21
> > **Acknowledging the Response**
> >
> > Thanks for the reply. I will wait for other reviewers to give their thoughts on the paper before deciding on my final score.

---

> > > ### Author Response · Authors · 2023-11-21
> > > **Official Comment by Authors**
> > >
> > > Sure, thank you for your time and effort!

---

### Official Review · Reviewer_BZux · 2023-10-31

**Soundness:** 3 good
**Presentation:** 3 good
**Contribution:** 2 fair
**Rating:** 6
**Confidence:** 4

**Summary:**

This paper proposes a novel parameter efficient tuning method for language models named decomposed prompt tuning or DePT. Compared with popular parameter efficient tuning method prompt tuning, deft aims to learn more compact soft prompt and a pair of low-rank matrices for updating the vocabulary embeddings. The key innovation for DePT is to offload the potential need for long soft prompts and decompose them into a pair of low rank matrices. Through experiments with both language models and vision-language models across various tasks, the authors find that DePT can achieve advantageous performance while saving approximately 20% memory cost.

**Strengths:**

To the best of the reviewer’s knowledge, the method proposed in this paper DeFT is novel. The authors also provide solid intuitions and reasoning for this method. Besides the method constructions, the experiments are comprehensive. I also appreciate the authors’ efforts in organizing the anonymous project code that covers the experiments.

**Weaknesses:**

The key contribution of DePT lies in that it is both optimizing a soft context as long as the vocabulary in an efficient manner. The decomposition idea, although novel in its current form, is incremental to current PEFT methods. There is also existing work (e.g. [1]) that explored tuning subsets of vocabularies as a way of PEFT. That being said, DePT still has the advantage of efficient vocabulary tuning. The 20% efficiency advantage also is only revealed with one soft prompt length of 100. It would be appreciated if the authors are to disclose more performance comparison under different prompt length scenarios.

Ref:
[1] Nayak, N. V., Yu, P., & Bach, S. (2022, September). Learning to Compose Soft Prompts for Compositional Zero-Shot Learning. In The Eleventh International Conference on Learning Representations.

**Questions:**

I appreciate the ablation experiments on the significance of different learning rates for the soft prompts and low rank matrices. I also wonder if there are other intuitive explanations.

---

> ### Author Response · Authors · 2023-11-16
> **Author Rebuttal by Authors**
>
> We appreciate the effort and time by the reviewer (BZux).
>
> `The key contribution of DePT lies in that it is both optimizing a soft context as long as the vocabulary in an efficient manner. The decomposition idea, although novel in its current form, is incremental to current PEFT methods. There is also existing work (e.g. [1]) that explored tuning subsets of vocabularies as a way of PEFT. That being said, DePT still has the advantage of efficient vocabulary tuning.`
>
> We thank the reviewer for the suggestion. We appreciate the opportunity to clarify: the meaning of “compose” and the method are fundamentally different between CSP [1] and our work:
>
> - **What is the difference?** CSP treats the attributes and objects that are composed to define classes as learnable tokens within the vocabulary. In contrast, our proposed method DePT does not train soft prompts associated with any vocabulary token, nor does it add additional tokens to the vocabulary. The main goal of DePT is to improve the efficiency of Prompt Tuning (PT) due to the increased input sequence. **DePT does not tune or update anything related to the vocabulary, including tokens and embeddings.**
> - **Which work is closely related to CSP and DePT respectively.** CSP [1] aligns more closely with prompt-based fine-tuning [2,3,4] where the soft prompts are trained associated with specific prompts and label words. In contrast, DePT aligns more closely with PT. While soft prompts in DePT and PT could be initialised with word embedding, these prompts are not updated in association with any existing or newly added vocabulary tokens. We will make this clear in our revised version.
> - **Comments from the reviewer (mz3i).** We appreciate that the reviewer (mz3i) has pointed out that while the title of our work is similar to the CSP work [1], the proposed methods are different. We have discussed this in the Appendix of our revised version.
>
> `The 20% efficiency advantage also is only revealed with one soft prompt length of 100. It would be appreciated if the authors are to disclose more performance comparison under different prompt length scenarios`
>
> We appreciate the reviewer's question. In response, we have performed additional experiments, as shown in the Table below. Specifically, we have increased the size of trainable parameters in both DePT and PT by a factor of two. We use the T5-base as the backbone. We report the average performance on the Glue dataset as well as inference speed, measured in inference samples per second. Our findings indicate that DePT (m=120, r=60) outperforms PT in terms of inference speed by 34%. We believe that this performance advantage can be further enhanced by reducing the value of m, which represents the length of the soft prompt. To provide a concrete example, on the SST-2 dataset, DePT can achieve an inference speed of 77.2 samples per second, while PT can only infer 57.4 samples per second. This suggests the advantage of DePT over PT increases as the model size increases.
>
> |                 | Mean Glue Performance  | Inference Speed  (Samples Per Second)|
> |-------     |-------|----|
> | DePT (m=120, r=60) | 86.0 |  54.8 |
> | PT (m=200) | 85.2 | 40.8 |
>
> (*m indicates the length of the soft prompt*, *r indicates the rank of low-rank matrices.*)
>
>
> `I appreciate the ablation experiments on the significance of different learning rates for the soft prompts and low-rank matrices. I also wonder if there are other intuitive explanations.`
>
> We thank the reviewer for the insightful suggestion. The intuition of our method is that (1) **given the same number of trainable parameters, allowing some updates for word embeddings will improve the performance**; and (2) **shorter soft prompt will improve the efficiency.** To illustrate, the previous study [5] has shown that a soft prompt can interpolate between many token embeddings, enabling the representation of more abstract concepts compared to relying on a single discrete token. However, the soft prompt in the Prompt Tuning is consistently added at the beginning of the frozen word embedding. In contrast, we propose DePT, which decomposes the long soft prompt into a short soft prompt and a pair of low-rank matrices. This approach can (1) reduce the length of the soft prompt for better efficiency; and (2) permit representation updates within the frozen word embedding, thereby increasing the adaptability of input representations that were previously unavailable.
>
> ### References:
>
> [1] Learning to Compose Soft Prompts for Compositional Zero-Shot Learning. ICLR 2023.
>
> [2] Exploiting cloze-questions for few-shot text classification and natural language inference. EACL 2021.
>
> [3] Differentiable prompt makes pre-trained language models better few-shot learners. ICLR 2022.
>
> [4] Don’t Stop Pretraining? Make Prompt-based Fine-tuning Powerful Learner, NeurIPS 2023.
>
> [5] Prompt Compression and Contrastive Conditioning for Controllability and Toxicity Reduction in Language Models. Findings of EMNLP 2022.

---

### Official Review · Reviewer_yLsP · 2023-11-02

**Soundness:** 3 good
**Presentation:** 3 good
**Contribution:** 3 good
**Rating:** 6
**Confidence:** 3

**Summary:**

The paper proposes a new PEFT method that falls under the prompt tuning category. The paper is motivated by the fact that soft prompts increase the sequence length which results in increased training and inference time. The main idea of the paper is to reduce the number of soft prompt tokens and use the remaining parameters to perform low-rank updates to the embedding matrix. The DePT method outperforms vanilla prompt tuning in almost all cases and is competitive with many other PEFT methods.

**Strengths:**

S1: The idea is very simple and leads to decent improvements over the baseline methods. Also, the paper is very easy to read and understand.

S2: The experiments are thorough enough, however, I have some mild additional suggestions that might make the experimental section more complete.

**Weaknesses:**

W1: Some of the important baseline methods like IA3 are missing. See questions below.

W2: The idea is interesting, however some more intuition on why this works might strengthen the paper.

**Questions:**

**For me to retain my current score**

Q: Many of the tables have inconsistent baseline and incomplete results. All the superglue results should be filled in Table 1. I understand that the numbers are not available in past papers but it is not hard to obtain these numbers. This is important because the LORA method is the closet peft method on the glue benchmark however it is missing when looking at Superglue. Similarly, Table 2 should have LoRA as a baseline.

Q: IA3 is a recent PEFT method that is very parameter efficient and is missing from the baseline comparison. It is a crucial baseline, especially in the case of the few-shot adaptation. It should be added at least to Tables 4 and 5 and adding it to Tables 1-2 would also be good.



**Other Questions**

Q: Most of the improvement in the table-1 glue task seems to come from 1-2 tasks like cola, rte any specific reason for this?


Note for other reviewers and AC: I am not sure of the related work section, there might be some relevant prompting-related baselines that I am not aware of.

---

> ### Author Response · Authors · 2023-11-16
> **Author Rebuttal by Authors**
>
> We appreciate the effort and time spent by the reviewer (yLsP).
>
> `W1: Some of the important baseline methods like IA3 are missing. See questions below.`
>
> We thank the reviewer for the insightful suggestion. In response, we have performed additional experiments as follows:
>
> (1) Firstly, we have added LoRA and IA3 as baselines and filled in the missing numbers for Table 1. Specifically, we train LoRA and IA3 with 30k steps with a batch size of 16. We use the rank as 35 for LoRA. We perform a learning rate search among 3e-3, 5e-4, 1e-4, and 5e-5, and report the test results of the best model on the development set. Our experiment shows that DePT obtains better mean performance on Glue and SuperGlue benchmarks.
>
> |                 | MNLI | QQP | QNLI | SST-2 | STS-B | MRPC | RTE | CoLA | Mean |
> |----------|-------|-------|----|----|------|------|------|------|------|
> |LoRA        | 86.3  | 89.0  | 93.2  | 94.3  | 90.9  | 90.1  | 75.5  | 63.3  | 85.3 |
> |IA3           |  85.7|  90.3 | 93.6 | 94.2 |   90.5 | 86.8 | 82.0  | 58.3 | 85.2 |
>
> |                                | Multirc | Bool   | WiC   | WSC | CB    | Mean |
> |----------------    |-------  |-------|----    |----    |------|------|
> |LoRA                       | 72.6     | 81.3    |  68.3 | 67.3   | 89.3  | 75.8 |
> |IA3                           | 73.3    | 80.9    |  67.7 | 73.1    | 85.7  | 76.1  |
> |Fine-tuning(m)       |  74.4     | 81.1     | 70.0  | 71.2    |  85.7 | 76.1  |
> |Adapter(m)             | 72.6  | 82.3     |  66.5 | 67.3   |  89.3  |  75.6 |
> |HyperFormer(m)    | 72.9  | 82.5      |  69.0 | 67.3   | 85.7  | 75.4 |
> |HyperDecoder(m)  | 70.4 |  78.8     | 67.1    | 61.5   | 82.1  | 72.0 |
> *m indicates multi-task learning*
>
> (2) We also add LoRA as a baseline in Table 2. While it is worth noting that LoRA performs highly competitively on WinoGrande, YelpPolarity, SciTail, and PAWS datasets, we have encountered challenges when training LoRA on certain MRQA tasks, such as HotpotQA. Our experiment shows that the advantage of DePT persists.
>
> |                 | NQ        | HP      | SQA  | News | Mean |
> |----------|-------  |-------|----    |----   |------  |
> |LoRA        | 69.7      |  61.6  |71.9   | 56.6  |    65.0 |
> |DePT   | 73.2 | 76.8 | 77.6 | 64.4 | 73.0 |
>
> | |  WG | Yelp      | SciTail | PAWS | Mean |
> |----------|-------  |-------|----      |----   |----   |
> |LoRA        | 58.2 | 97.1       | 94.7   | 94.0    | 86.0 |
> |DePT   | 59.0 | 96.8 | 95.6 | 93.7 | 86.3 |
>
> (3) In addition, we have added IA3 as a baseline to Table 4 in the few-shot learning setting (4, 16, or 32 training examples). We report average results across three seeds. Specifically, we find that DePT, along with other PT variants (ATTEMPT and MPT), can outperform IA3, highlighting the effectiveness of DePT with transfer learning approaches.
>
> |                 | 4  | 16  | 32  |
> |-------     |-------|----|----   |
> | Boolq      | 56.7  | 62.0 | 67.2 |
> | CB          | 65.5 | 71.4 | 75.0 |
> | SciTail      | 65.4 | 74.4 | 80.4 |
>
> `W2: The idea is interesting, however some more intuition on why this works might strengthen the paper.`
>
> We acknowledge the reviewer's inquiry regarding the intuition of our method. The intuition of our method is that (1) **given the same number of trainable parameters, allowing some updates for word embeddings will improve the performance**; and (2) **shorter soft prompt will improve the efficiency.** To illustrate, the previous study [1] has shown that a soft prompt can interpolate between many token embeddings, enabling the representation of more abstract concepts compared to relying on a single discrete token. However, the soft prompt in the Prompt Tuning is consistently added at the beginning of the frozen word embedding. In contrast, we propose DePT, which decomposes the long soft prompt into a short soft prompt and a pair of low-rank matrices. This approach can (1) reduce the length of the soft prompt for better efficiency; and (2) permit representation updates within the frozen word embedding, thereby increasing the adaptability of input representations that were previously unavailable.
>
> [1] Prompt Compression and Contrastive Conditioning for Controllability and Toxicity Reduction in Language Models. Findings of EMNLP 2022.
>
> `Q: Most of the improvement in the table-1 glue task seems to come from 1-2 tasks like cola, rte any specific reason for this?`
>
> We appreciate the reviewer's question. Our main objective in this study is not solely to improve performance metrics in the table. Instead, we seek to demonstrate that DePT can deliver highly competitive or even better overall performance while substantially enhancing overall efficacy in terms of computational time and memory usage, which is especially useful for large language models.

---

> > ### Comment · Reviewer_yLsP · 2023-11-21
> > **Response**
> >
> > I thank the authors for their reply.  I have read the reply and will retain my score.
> >
> > Just a small question, in table 2 the performance of lora for mrqa seem really low I would suggest the authors to recheck their implementation and tuning it more for the final version.

---

> > > ### Author Response · Authors · 2023-11-21
> > > **Author Rebuttal by Authors**
> > >
> > > Thank you so much for your continued discussion and feedback!
> > >
> > > We fully agree on this point: the problem probably comes from not having enough time to conduct an exhaustive hyperparameter search. We will keep working on this and ensure to update the score in the final version of the paper.

---

### Author Response · Authors · 2023-11-16
**Author Rebuttal by Authors**

Thank you to all the reviewers for dedicating their time and effort to evaluate our work.  We are thrilled to receive positive feedback on the novelty of our approach (`BZux`), the simplicity and effectiveness of our approach (`yLsP`), solid intuitions and reasoning (`BZux`), solid experimental evidence (`yLsP`,`BZux`,`XBEs`), the quality of the presentation (`yLsP`,`mz3i`,`XBEs`), and well-organised code (`BZux`,`mz3i`).

To answer the reviewers’ questions, we have conducted multiple additional experiments, regarding `adding LoRA and IA3 as baselines in Table 1, 2 and 4`, `filling in missing baselines from previous works`, `evaluating DePT with larger model sizes`, and `evaluating DePT with different prompt lengths`. We have incorporated these changes in the main text and the Appendix of our paper (marked in red). Given the constraints on space, we have placed the most of additional content in the Appendix. We hope that our response, paired with these additional experiments, will address the reviewers' concerns.

---

### Meta-Review · Area_Chair_onTw · 2023-12-08

**Metareview:**

This paper proposes to improve prompt tuning by learning a shorter soft prompt and a low-rank update to the model's word embeddings. The goal is to shorten the number of prompt tokens (which increase memory and computation) while retaining the total number of updated parameters compared to prompt tuning and ultimately improving performance. Sure enough, the method does just that. Overall, the results are solid and the method does meaningfully improve over prompt tuning on a rather wide set of experiments. In addition, it outperforms various baseline PEFT methods. If anything, one weakness could be that the method is a pretty simple modification to prompt tuning, where the modification is basically doing LoRA on the embedding matrix. Reviewers also pointed out many missing baseline experiments, but the authors addressed these issues. All reviewers therefore agreed on acceptance.

**Justification For Why Not Higher Score:**

While DePT presents a clear improvement over PT, it's a pretty incremental change.

**Justification For Why Not Lower Score:**

All reviewers recommended acceptance.

---

### Decision · Program_Chairs · 2024-01-16

Accept (poster)